

# Evaluation of four calving laws for Antarctic ice shelves

Joel A. Wilner[1], Mathieu Morlighem[1], and Gong Cheng[1]

[1]Department of Earth Sciences, Dartmouth College, Hanover, NH, USA

**Correspondence:** Joel A. Wilner (joel.a.wilner.gr@dartmouth.edu)

**Abstract.** Many floating ice shelves in Antarctica buttress the ice streams feeding them, thereby reducing the discharge of icebergs into the ocean. The rate at which ice shelves calve icebergs and how fast they flow determines whether they advance, retreat, or remain stable, exerting a first-order control on ice discharge. To parameterize calving within ice sheet models, several empirical and physical calving "laws" have been proposed in the past few decades. Such laws emphasize dissimilar
features, including along- and across-flow strain rates (the eigencalving law), a fracture yield criterion (the von Mises law), longitudinal stretching (the crevasse depth law), and a simple ice thickness threshold (the minimum thickness law), among others. Despite the multitude of established calving laws, these laws remain largely unvalidated for the Antarctic Ice Sheet, rendering it difficult to assess the broad applicability of any given law in Antarctica. We address this shortcoming through a set of numerical experiments that evaluate existing calving laws for ten ice shelves around the Antarctic Ice Sheet. We utilize the
Ice-sheet and Sea-level System Model (ISSM) and implement four calving laws under constant external forcing, calibrating the free parameter of each of these calving laws by assuming that the current position of the ice front is in steady state and finding the set of parameters that best achieves this position over a simulation of 200 years. We find that, in general, the eigencalving and von Mises laws best reproduce observed calving front positions under the steady state position assumption. These results will streamline future modeling efforts of Antarctic ice shelves by better informing the relevant physics of Antarctic-style
calving on a shelf-by-shelf basis.

## 1 Introduction

Iceberg calving is a major component of the mass budget of large ice sheets and marine-terminating glaciers. Currently, calving constitutes approximately 45% of mass loss from Antarctic ice shelves (Rignot et al., 2013) and, together with frontal melt,
approximately 40% from the Greenland Ice Sheet (Aschwanden et al., 2019). Accelerated ice discharge through calving amplifies the mass deficit of ice sheets and ice caps, enhancing their contribution to global sea-level rise (Benn et al., 2007b). This contribution is disproportionately large given that calving is highly responsive to initial climate signals, which precipitate rapid glacier retreat (Meier and Post, 1987; Rignot et al., 2003; Benn et al., 2007a). Additionally, by reducing resistive stresses at the glacier base and margins, enhanced calving leads to upstream thinning and acceleration, initiating a strong positive feedback



on glacier dynamics that further accelerates ice discharge (e.g., Gagliardini et al., 2010; Choi et al., 2018). Given these dynamic feedbacks, realistic simulation of calving in large-scale ice sheet models is crucial for sea-level rise projections over the course of the 21st century and beyond (e.g., Yu et al., 2017; Amaral et al., 2020).

Despite its central role in projecting sea-level rise, iceberg calving remains a fundamental challenge of ice sheet modeling. Several calving laws have been proposed over the past few decades (e.g., Hughes, 1992; Levermann et al., 2012; Morlighem

et al., 2016; Mercenier et al., 2018), but a universal calving law – a reliable predictive mathematical formula for calving – has eluded description and consensus. Proposed calving laws are typically limited by the inherent physical complexity and vast diversity of observed calving processes (Benn et al., 2007b; Bassis and Jacobs, 2013; Benn et al., 2017). These processes range from the consistent detachment of small-scale icebergs from fjord-confined outlet glaciers (typical of Greenland) to the sporadic calving of massive tabular icebergs off of ice shelves (typical of Antarctica) (Vieli et al., 2001; Bassis, 2011). This

discrepancy is primarily a function of the predominance of grounded tidewater glaciers in Greenland versus large floating ice shelves in Antarctica. Examples of these respective calving regimes include the well-developed mélange of calving discharge at Jakobshavn Isbræ, Greenland (Amundson et al., 2010; Cassotto et al., 2015) and the large concentrations of tabular icebergs in the Southern Atlantic and Southern Indian Oceans (Tournadre et al., 2012). A satisfactory calving law would be able to describe the processes responsible for Greenland-style and Antarctica-style calving along a mathematical continuum.

To parameterize such processes, most numerical models of the Antarctic and Greenland Ice Sheets utilize simple empirical calving laws as boundary criteria, classified into *position* laws and *rate* laws. The former express calving in terms of terminus position change, whereas the latter express calving in terms of iceberg calving rate. Accordingly, position laws parameterize calving at discrete intervals, whereas rate laws parameterize calving continuously. These laws are typically constrained by satellite-derived measurements or inferences of ice velocity, ice stresses, and fjord geometry, among other constraints. Early

efforts at calving parameterization expressed calving rate in terms of water depth (e.g., Brown et al., 1983) or forced terminus thickness to exceed a value controlled by some calving criterion (e.g., Van Der Veen, 1996; Pfeffer et al., 1997). However, these laws had difficulty applying to both glaciers with grounded and floating termini (Amundson and Truffer, 2010). The crevasse depth law of Benn et al. (2007b) was one of the first to overcome this difficulty, predicting calving to occur when the depth of surface crevasses equals the freeboard (i.e., ice surface height above water level). In this scenario, calving is

effectively governed by longitudinal strain rate, which exerts a first-order control on crevasse depth. Despite the success of a strain-based law at bridging the gap between grounded and floating termini, inconsistencies persisted when describing a range of calving behaviors, prompting the development of new calving laws (Amundson and Truffer, 2010). Examples of such laws include an extension of the crevasse depth law that incorporates propagation of basal crevasses (Nick et al., 2010); an application of the crevasse depth law to floating ice shelves (CD; Pollard et al., 2015); the eigencalving law (EC; Levermann

et al., 2012), an expression of calving rate as proportional to the product of strain rates parallel and perpendicular to flow; the von Mises law (VM; Morlighem et al., 2016), reliant on tensile stresses and frontal velocities; and a simple minimum thickness threshold (MT). In contrast to these semi-empirical laws, a class of laws has emerged that describe calving through predominantly physical means. Such laws incorporate a continuum damage framework (e.g., Duddu et al., 2013), linear elastic fracture mechanics (Yu et al., 2017), or a combination of the two (Krug et al., 2014).



The development of increasingly complex calving laws has outpaced law validation, leading to significant uncertainty in the applicability of calving laws at the ice-sheet scale (Benn et al., 2017; Amaral et al., 2020). Where data are readily available, calving laws tend to be tuned to specific glaciers or regions, failing to generalize for a range of calving behaviors (Benn et al., 2007b; Bassis, 2011). To overcome this problem, two studies have systematically compared calving laws between different outlet glaciers in Greenland (Choi et al., 2018; Amaral et al., 2020). Choi et al. (2018) simulated glacier flow from 2007 to 2017 with five different calving laws: height above buoyancy (Vieli et al., 2001), the crevasse depth laws of Benn et al. (2007b) and Nick et al. (2010), EC, and VM. In general, VM was found to most closely reproduce observed terminus position change of the nine Greenland outlet glaciers under consideration, with EC performing the poorest. Notably, the authors cautioned that the discharge of large tabular icebergs in Antarctica is likely governed by different physical principles so the results cannot be considered universal. Amaral et al. (2020) evaluated three position laws (the crevasse depth law of Nick et al. (2010), height above flotation, and fraction above flotation) and three rate laws (the stress maximum law of Mercenier et al. (2018), EC, and VM), comparing flowline dynamics against spatially and temporally diverse observations of 50 tidewater glaciers in Greenland. Unlike Choi et al. (2018), Amaral et al. (2020) concluded that the crevasse depth law of Nick et al. (2010) captures observed terminus dynamics with high fidelity. Moreover, Amaral et al. (2020) found that rate-based laws tend to produce high misfit values, particularly for large ablation rates, and struggle to reproduce seasonal and short-term variability. The disparate results of Choi et al. (2018) and Amaral et al. (2020) may be attributed to their contrasting modeling strategies, as the former study utilized the Ice-sheet and Sea-level System Model (ISSM; Larour et al., 2012) with the 2-D Shelfy-Stream Approximation (SSA; Morland and Zainuddin, 1987; MacAyeal, 1989) and a level-set method (Bondzio et al., 2016) to track calving front positional changes whereas the latter study employed a simpler 1-D flowline approach. Another important distinction is that Choi et al. (2018) calibrated calving law tuning parameters on a glacier-specific basis, whereas Amaral et al. (2020) made use of a uniform tuning parameter value across a more extensive suite of glaciers.

To date, calving laws have not been systematically compared for Antarctic ice shelves, rendering it difficult to assess the broad applicability of any given calving law in simulations of Antarctic Ice Sheet dynamics. Here, we follow the approach of Choi et al. (2018) in comparing calving laws for Antarctica, reasoning that the SSA is well suited to model Antarctic ice shelves and that calibrating tuning parameters specific to each ice shelf is a necessary step to optimize calving law performance. However, unlike in Greenland, where calving events are frequent and there are abundant temporal changes in front positions to compare models against, calving events in Antarctica are typically episodic, in some instances only occurring once per several decades for a given portion of the ice margin (Fricker et al., 2002). For this reason, we conclude that it would be impractical to compare modeled front positions for Antarctica against a series of observed changes in front position. Thus, a key difference between our methodology and that of Choi et al. (2018) is that we instead assume that the modern front position is in steady state and try to attain the modeled front that most closely replicates this one position over a 200-year run under constant forcing.

We ascertain that calving processes at different ice shelves may be governed by dissimilar physical principles as a consequence of varying degrees of lateral confinement (among many other potential factors), and that unique tuning parameter values must be determined accordingly. To this end, we implement four different calving laws in ISSM: EC, VM, MT, and CD. We apply these calving laws to ten Antarctic ice shelves, selected on the basis of their size or abundance of observational



data. Each calving law has one or two free tuning parameters, calibrated by running 200-year forward models of each ice shelf and determining which parameter value most closely fits the modern-day observed calving front position under a steady state assumption. We then compare the four calibrated calving laws to each other for each ice shelf, determining which calving law best captures a steady state front position on a shelf-by-shelf basis. We quantify possible correlates to calving law success, including frontal velocity, thickness, and buttressing force. Finally, we discuss the implications of these results for ice sheet projections in a warming climate.

## 2 Methods

Like Choi et al. (2018), we use ISSM to implement four calving laws (EC, VM, MT, and CD) for ten ice shelves: Amery, Denman, Filchner, Larsen C, Pine Island, Ronne, Ross, Shackleton, Thwaites, and Totten Ice Shelves. Mesh resolution increases with proximity to the observed calving front, with a highest resolution of 1 km near the calving front, and is isotropically adapted according to ice velocity. Larger ice shelves such as Ross and Ronne are prescribed a coarser maximum resolution (3 km) for computational efficiency. Typical mesh size for all ice shelves is on the order of tens of thousands of elements. We find that increasing the mesh resolution further does not influence our main results in any significant way. Surface elevation and bed topography data are taken from BedMachine Antarctica V3 (Morlighem et al., 2020). Observed ice velocity data are retrieved from the NASA Making Earth System Data Records for Use in Research Environments (MEaSUREs) dataset, which uses satellite-derived, phase-based InSAR and tracking measurements to derive ice velocity over the period of 2007 to 2018 (Mouginot et al., 2019). As in Choi et al. (2018), surface mass balance (SMB) input is kept constant over the simulation period and is derived from the Regional Atmospheric Climate Model (RACMO2.3) (Noël et al., 2015). Our ice shelf models are initialized by first inverting for rigidity over floating regions of the domain followed by inverting for basal friction in grounded regions. To reiterate, we seek the modeled calving front that most closely matches the shape and position of the modern observed front, assumed to be in steady state because calving in Antarctica occurs so infrequently, unlike in Greenland. We set the transient model to run for an arbitrarily long time of 200 years, which we deem long enough to attain a modeled steady state calving front position for calibrated tuning parameters.

### 2.1 Calving laws

Three of the four calving laws used here are rate-based laws (EC, VM, and CD), whereas MT is a position-based law. EC defines calving rate $c$ as proportional to along- and across-flow strain rate, such that

$$c = K \, \dot{\epsilon}_{\parallel} \, \dot{\epsilon}_{\perp}, \tag{1}$$

where $K$ is a proportionality constant that encapsulates the material properties of ice relevant to calving, $\dot{\epsilon}_{\parallel}$ is along-flow strain rate, and $\dot{\epsilon}_{\perp}$ is across-flow strain rate (Levermann et al., 2012). By definition, the EC law contains a first-order dependence on the spreading rate tensor and thus may be suited to capture calving behavior of ice shelves in unconfined embayments. We take $K$ to be the tuning parameter for EC in our calibration experiments.





Within VM, $c$ is proportional to the tensile stress regime at the ice front. Mathematically, this is represented as

$$c = |\boldsymbol{v}| \frac{\tilde{\sigma}}{\sigma_{\max}}, \tag{2}$$

where $\boldsymbol{v}$ is the ice front velocity vector, $\tilde{\sigma}$ is the total tensile von Mises stress at the ice front, and $\sigma_{\max}$ is the tensile stress threshold (taken to be the tuning parameter for VM). Without considering any additional undercutting at the ice front, retreat
occurs when $\tilde{\sigma}$ exceeds $\sigma_{\max}$, and advance occurs when $\tilde{\sigma}$ is less than $\sigma_{\max}$. Total von Mises tensile stress $\tilde{\sigma}$ is defined as

$$\tilde{\sigma} = \sqrt{3} B \tilde{\dot{\varepsilon}}_e^{1/n}, \tag{3}$$

where $B$ is the ice viscosity parameter, $n$ is Glen's law exponent (taken here to be 3), and $\tilde{\dot{\varepsilon}}_e$ is the effective tensile strain rate, itself proportional to the eigenvalues of the 2-D horizontal strain rate tensor $\dot{\varepsilon}_1$ and $\dot{\varepsilon}_2$ (Morlighem et al., 2016):

$$\tilde{\dot{\varepsilon}}_e = \frac{1}{2} \left( \max(0, \dot{\varepsilon}_1)^2 + \max(0, \dot{\varepsilon}_2)^2 \right). \tag{4}$$

The third and final rate law considered here, CD, is a modification of the ice shelf calving criterion introduced by Pollard et al. (2015). Although the crevasse depth laws of Benn et al. (2007b) and Nick et al. (2010) do allow for the development of floating ice tongues, those parameterizations are not initially tuned for large floating ice shelves. In addition, a satisfactory crevasse depth law for floating ice shelves would ideally account for such factors as the discrepancy in ice velocity at the grounding line versus shelf interior and edge points, as in Pollard et al. (2015). Their initial parameterization of incipient dry-surface and
basal crevasse depths $d_s$ and $d_b$, respectively, follows Nick et al. (2010). Dry-surface crevasse depth is parameterized as

$$d_s = \frac{2}{\rho_i g} \left( \frac{\nabla \cdot \boldsymbol{v}}{A} \right)^{\frac{1}{n}}, \tag{5}$$

where $\nabla \cdot \boldsymbol{v}$ is ice divergence, $A$ is the depth-averaged ice rheological coefficient, $g$ is gravitational acceleration, and $\rho_i$ is ice density. Basal crevasse depth is parameterized as

$$d_b = \left( \frac{\rho_i}{\rho_w - \rho_i} \right) \frac{2}{\rho_i g} \left( \frac{\nabla \cdot \boldsymbol{v}}{A} \right)^{\frac{1}{n}}, \tag{6}$$

where $\rho_w$ is the density of ocean water. Pollard et al. (2015) include a velocity dependency, parameterizing additional crevasse deepening $d_a$ of incipient fractures as

$$d_a = h \max[0, \ln(|\boldsymbol{v}|/1600)]/\ln(1.2) \tag{7}$$

where $h$ is the local ice thickness and $\boldsymbol{v}$ is here expressed in units of m yr$^{-1}$. The constants in this equation are values calibrated by Pollard et al. (2015) to match velocity observations of Ross and Ronne Ice Shelves as case examples. That is, $d_a$ is tuned to
approach $h$ as local ice speed increases to $\sim$1900 m yr$^{-1}$, speeds observed in the fast-flowing outer regions of Ross and Ronne Ice Shelves. For speeds up to 1600 m yr$^{-1}$, $d_a$ is zero. A crevasse depth component $d_t$ which accounts for thin floating ice is introduced:

$$d_t = h \max[0, \min[1, (150 - h)/50]]. \tag{8}$$



This equation has the practical effect of removing unrealistically thin floating ice less than 100 to 150 m thick. Although this
prevents small tidewater glaciers from extending into ice shelves, such small glaciers are excluded from our study. A final
crevasse depth component $d_w$ is imposed by Pollard et al. (2015), accounting for the additional opening stress of liquid water
in surface crevasses:

$$d_w = 100\,R^2 \tag{9}$$

where $R$ is annual surface melt plus liquid rainfall remaining after refreezing (in m yr$^{-1}$). We account for this term through
the constant SMB derived from RACMO2.3. Taken together, these individual crevasse depth components can be expressed as
a ratio to ice thickness:

$$r = \frac{d_s + d_b + d_a + d_t + d_w}{h}, \tag{10}$$

and finally an overall calving rate (here in m yr$^{-1}$) is obtained:

$$c = \dot{M}_{\max}\,\max\left(0, \min\left(1, \frac{r - r_c}{1 - r_c}\right)\right), \tag{11}$$

where $\dot{M}_{\max}$ is the maximum migration rate of the ice front (in m yr$^{-1}$) and $r_c$ is some critical value for calving onset between
0 and 1. We take $\dot{M}_{\max}$ and $r_c$ as the tuning parameters for the CD law. Note that we cap $\dot{M}_{\max}$ at 10 km yr$^{-1}$ for all calving
laws to prohibit unrealistically large advance or retreat, but since $\dot{M}_{\max}$ is a tuning parameter for CD, we vary it between 2 km
yr$^{-1}$ and 10 km yr$^{-1}$ during CD calibration.

The MT law, which differs from the other three laws considered here in that it is a position law, simply states that calving
occurs when $h \le h_{\min}$, where $h_{\min}$ is some minimum ice thickness threshold, taken here as a tuning parameter.

## 2.2 Calibration procedure

To calibrate the four calving laws, we perform a series of tests to compute a misfit metric for each ice shelf. First, we calculate
the area of the region between the observed modern-day ice shelf front and the modeled front after the simulation period of
200 years. We then divide this area by the length of the observed modern-day front to obtain a misfit (in km). Note that this
definition does not consider the sign of the misfit; the misfit is agnostic to retreat or advance. A key assumption of this misfit
calculation is that the ice shelf front is in a steady state position. This assumption generally holds true for the ice shelves under
consideration, which calve sporadically enough that their front positions over the past few decades have been roughly constant.
A major exception to this assumption is Thwaites Ice Shelf, whose front is much more dynamic than that of other ice shelves
under consideration, but we have included it in this study for the sake of completeness and its importance to the stability of the
West Antarctic Ice Sheet.

We compute the misfit metric for a range of values of each calving law's tuning parameter(s), taking the best-fit as the value
that minimizes the misfit (Fig. 1). In constraining the precision of the calibration, we vary the tuning parameter at intervals of
5 kPa for $\sigma_{\max}$ of VM, 5 m for $h_{\min}$ of MT, and 0.1 and 2 km for $r_c$ and $\dot{M}_{\max}$, respectively, of CD. The calibration precision
for $K$ of EC varies by ice shelf: $0.5 \times 10^7$ m×yr for Filchner, Amery, and Denman, and $0.5 \times 10^8$ m×yr for all other ice shelves

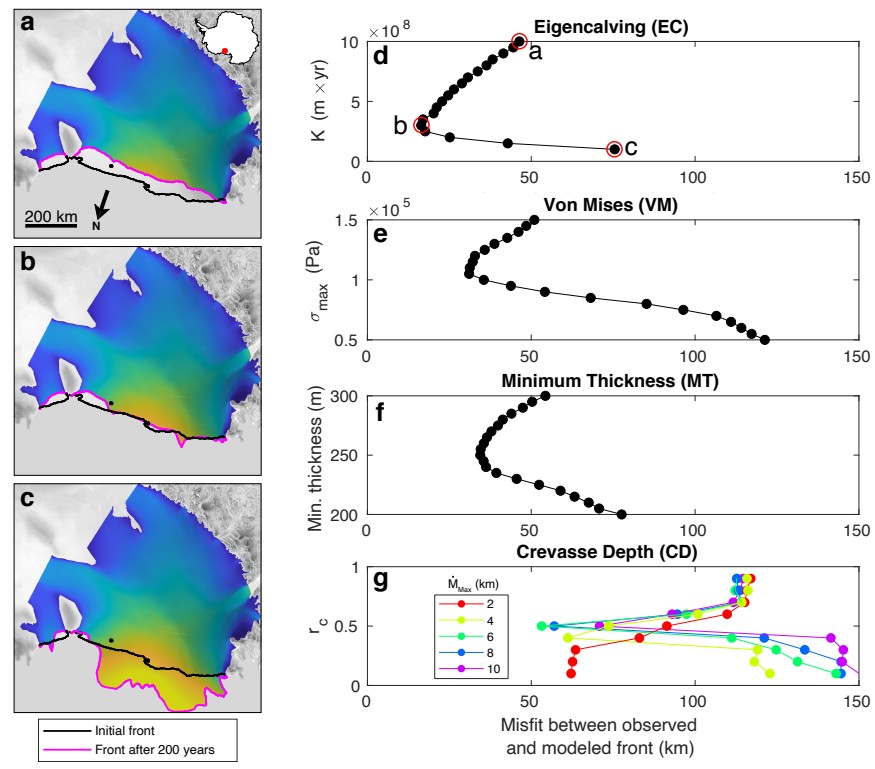

**Figure 1.** Example of the calibration process for Ross Ice Shelf. Representative results showing **(a)** retreat, **(b)** steady state, and **(c)** advance for eigencalving are shown on the left along with ice velocity in $\log_{10}$ scale from 1 m/yr (blue) to 1000 m/yr (yellow). Calibration curves for the tuning parameters of **(d)** EC, **(e)** VM, **(f)** MT, and **(g)** CD are shown on the right for misfit distances between 0 and 150 km. EC tuning parameter values corresponding to model scenarios **(a)**, **(b)**, and **(c)** are circled in panel **(d)**. Model results are overlaid with a radar image in grey, and the location of Ross Ice Shelf is shown by a red dot on an Antarctica inset.

except Larsen C ($0.5 \times 10^{10}$ m×yr). The typical outcome of the calibration process is a curve with one branch representing retreat from the modern-day observed front and another branch representing advance, as shown in Figs. 1d-g for Ross Ice Shelf. Fig. 1a and Fig. 1c show the most extreme retreat and advance scenarios, respectively, modeled for EC. The optimal steady state (Fig. 1b) occurs for a tuning parameter value where the misfit metric is minimized on the calibration curve.

## 3  Results

Tuning parameter calibration results are shown in Table 1. Notably, the chosen $K$ values of EC range 4 orders of magnitude, with Larsen C being an upper outlier. The chosen $\sigma_{\mathrm{max}}$ values of VM are generally consistent with each other (same order of magnitude), while the chosen $h_{\mathrm{min}}$ values of MT are largely dependent on the original thickness of the ice shelf. The chosen CD parameter value $r_c$ ranges greatly between 0 and 1, signifying a diversity of critical crevasse depth thresholds, while the



| Ice Shelf | Calving calibration parameter | | | | |
|---|---|---|---|---|---|
| | $K$ of EC $\times 10^8$ (m×yr) | $\sigma_{\mathrm{max}}$ of VM (kPa) | $h_{\mathrm{min}}$ of MT (m) | $r_c$ of CD (unitless) | $\dot{M}_{\mathrm{max}}$ of CD (km) |
| Amery | 0.50 | 150 | 235 | 0.4 | 2.0 |
| Denman | 0.20 | 400 | 215 | 0.9 | 2.0 |
| Filchner | 1.5 | 200 | 440 | 0.5 | 8.0 |
| Larsen C | 300 | 140 | 230 | 0.5 | 2.0 |
| Pine Island | 3.0 | 275 | 390 | 0.9 | 2.0 |
| Ronne | 4.0 | 120 | 280 | 0.5 | 6.0 |
| Ross | 3.0 | 105 | 250 | 0.5 | 6.0 |
| Shackleton | 2.0 | 280 | 55 | 0.9 | 2.0 |
| Thwaites | 2.0 | 320 | 395 | 0.1 | 2.0 |
| Totten | 2.5 | 260 | 410 | 0.5 | 2.0 |

**Table 1.** Tuned calibration parameter values that minimize the misfit metric, by ice shelf. CD parameter values ($r_c$ and $\dot{M}_{\mathrm{max}}$) are to be taken in combination.

corresponding CD parameter $\dot{M}_{\mathrm{max}}$ is typically minimized at 2 km except for the three largest ice shelves, Ross, Ronne and Filchner.

The calibration results visualized for Ross Ice Shelf represent an ideal scenario where one calving law (EC) performs markedly better than the others at minimizing misfit between initial and modeled fronts (Fig. 1d-g). In this example, EC closely attains the shape and position of the assumed steady state observed calving front (Fig. 1b). As such, in ascertaining calving law success, it is instructive to consider not only the quantitative misfit metric but to also qualitatively assess how well the modeled front captures the shape and position of the observed front.

We now present results for all ice shelves under consideration, roughly grouped by region. Only one calving law, VM, captures the modern-day calving front position of Larsen C Ice Shelf with any fidelity (Fig. 2a). Interestingly, EC results in considerable ice advance in the region immediately adjacent to Gipps Ice Rise and downstream of the Kenyon Peninsula. The spatial distribution of this advance stands in contrast to the notion that ice flow becomes less constrained by the lateral shear margins after passing Kenyon Peninsula (Wang et al., 2022), since regions of decreased lateral confinement, and thus enhanced lateral strain rate, are expected to increase calving rates under the EC framework (Eq. 1). Both MT and CD result in considerable retreat in the southern portion of the ice shelf, with MT exhibiting pronounced advance in the central portion. Notably, the divide between advance and retreat produced by MT closely follows the arcuate downstream profile of ice extending from Kenyon Peninsula. Qualitatively, misfit is consistently minimized by VM across the extent of the ice shelf front, and the associated misfit value of 6.0 km is more than twice as favorable as the next best misfit value (EC; 12.7 km). CD (22.1 km) and MT (35.8 km) have markedly larger misfit values for Larsen C. Ronne Ice Shelf is best fit by EC (5.7 km), followed by VM (18.2 km), MT (23.8 km), and CD (29.0 km). EC and VM capture the shape and position of the front well except for a south-central region where EC results in a thin protruding ice tongue, coincident with a broader ice tongue associated with VM (Fig. 2b). MT and CD exhibit undulating front shapes which do not closely replicate Ronne's sublinear front. In contrast to Ronne, Filchner Ice





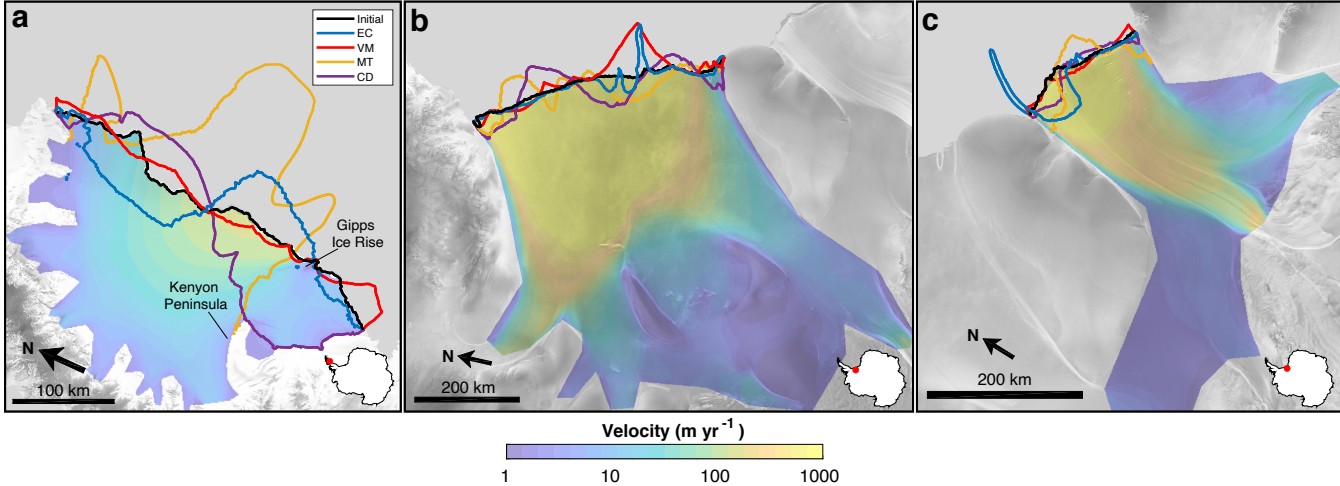

**Figure 2.** Model results for **(a)** Larsen C, **(b)** Ronne and **(c)** Filchner Ice Shelves. The modeled front positions for each calving law are shown in different colors, along with the initial observed front in black. Ice shelf location is shown by a red dot on an Antarctica inset. Model results are overlaid with a radar image in grey.

Shelf is best fit by VM (4.9 km), followed by CD (9.2 km), MT (21.1 km), and EC (28.9 km). Only VM adequately captures the western portion of the shelf; the other calving laws struggle to replicate the frontal shape and position, generating too much retreat and, in the case of EC, producing an extensive ice tongue at the western shelf margin (Fig. 2c).

Pine Island and Ross Ice Shelves are best fit by EC (both 16.4 km) and Thwaites is best fit by CD (5.7 km). For Pine Island, the next best calving laws are VM (20.1 km), MT (20.4 km), and CD (47.1 km); for Thwaites, the next best calving laws are EC (6.4 km), VM (7.4 km), and MT (7.8 km); and for Ross, the next best calving laws are VM (31.0 km), MT (34.4 km), and CD (53.2 km). All four calving laws, particularly CD, result in some level of frontal irregularity for Pine Island, though VM captures the northern portion of the front with high fidelity (Fig. 3a). None of the four calving laws replicate the observed ice tongues in the central portion of Thwaites, but CD and VM replicate the front position of the western portion of Thwaites with some success (Fig. 3b). With the exception of a small ice tongue produced in the western portion of the front and some retreat in the eastern portion, EC generally maintains the frontal shape and position of Ross Ice Shelf, in keeping with minimizing the misfit metric (Fig. 3c).

For Amery Ice Shelf, we observe that the best-fit model runs of EC and VM produce ice fronts inverse in shape to each other, with concave and convex fronts, respectively (Fig. 4a). The convex front of VM more closely aligns with the modern-day observed front that extends beyond the embayment under our assumption of steady state front position. Indeed, out of the four calving laws for Amery, VM minimizes the misfit metric (9.2 km), followed by MT (9.5 km), EC (17.3 km), and CD (21.2 km). MT performs reasonably well at capturing the shape and position of the front, particularly in the easternmost portion, though it overestimates advance for the westernmost half. The misfit values of VM and MT are nearly the same, whereas EC and CD perform markedly worse both in both quantitative and qualitative senses, with large regions of retreat with respect



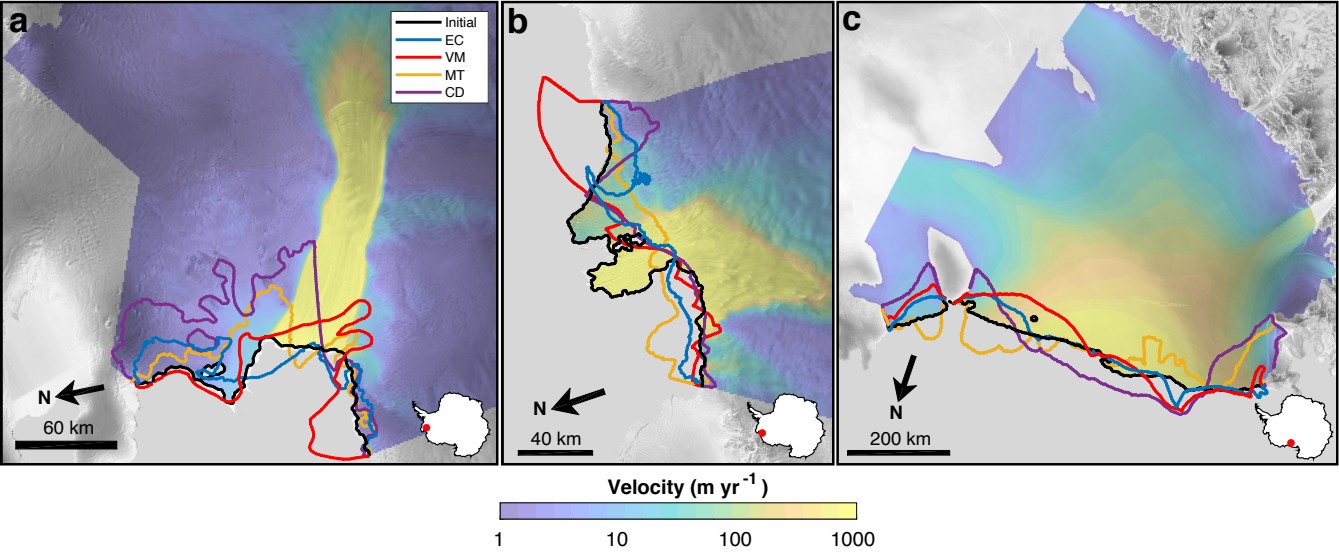

**Figure 3.** Model results for **(a)** Pine Island, **(b)** Thwaites and **(c)** Ross Ice Shelves. The modeled front positions for each calving law are shown in different colors, along with the initial observed front in black. Ice shelf location is shown by a red dot on an Antarctica inset. Model results are overlaid with a radar image in grey.

to the modern-day observed front. Shackleton Ice Shelf involves a moderate level of modeled frontal irregularity, though VM (3.5 km) and EC (3.7 km) capture the observed central protrusion of the ice front reasonably well (Fig. 4b). MT (6.1 km) and CD (8.2 km) truncate this protrusion. Modeled fronts for Denman Ice Shelf are irregularly shaped, with the best-fit law VM (3.5 km) replicating the easternmost portion of the front well but overestimating advance in the westernmost portion (Fig. 4c). EC (7.5 km) and MT (9.6 km) exhibit highly irregular patterns of retreat, and CD (13.1 km) produces a long and spurious ice tongue. All calving laws perform reasonably well for Totten Ice Shelf, with low misfit values of <5 km, though only EC (1.9 km) accurately captures the convexity of the ice front (Fig. 4d). MT (4.0 km), VM (4.1 km) and CD (4.5 km) all exhibit retreat in the convex portion of the ice shelf. EC does produce some retreat in the eastern region where MT and CD replicate the observed front, but we deem this region to be insignificant compared to the bulk of the calving front which does demonstrate affinity to EC.

The overall results indicate that 9 of the 10 ice shelves under consideration are best captured by either EC (Pine Island, Ronne, Ross, and Totten) or VM (Amery, Denman, Filchner, Larsen C, and Shackleton), assuming our steady state assumption of front position (Fig. 5). In some cases, such as VM versus MT for Amery and VM versus EC for Shackleton, the best-fit calving law is only marginally better than the second-best. MT results in several unacceptably large misfits (Filchner, Larsen C, Ronne, and Ross), as does CD (Pine Island, Ronne, and Ross). EC misfit is unacceptably large for Filchner, but maintains a low value for most other ice shelves, relative to the original size of the ice shelf. Only Thwaites Ice Shelf is best characterized by a calving law other than EC or VM (CD), though it is important to reiterate that Thwaites exhibits more dynamic behavior than the other ice shelves considered here.





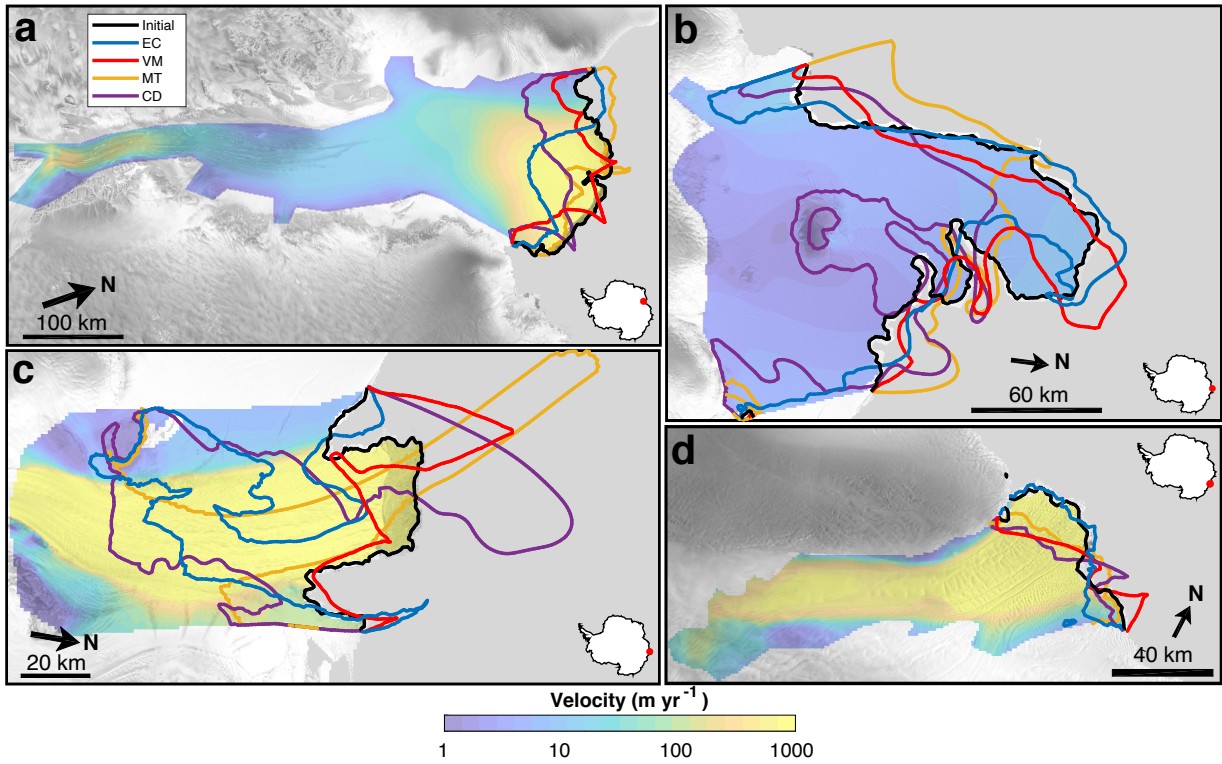

**Figure 4.** Model results for **(a)** Amery, **(b)** Shackleton, **(c)** Denman, and **(d)** Totten Ice Shelves. The modeled front positions for each calving law are shown in different colors, along with the initial observed front in black. Ice shelf location is shown by a red dot on an Antarctica inset. Model results are overlaid with a radar image in grey.

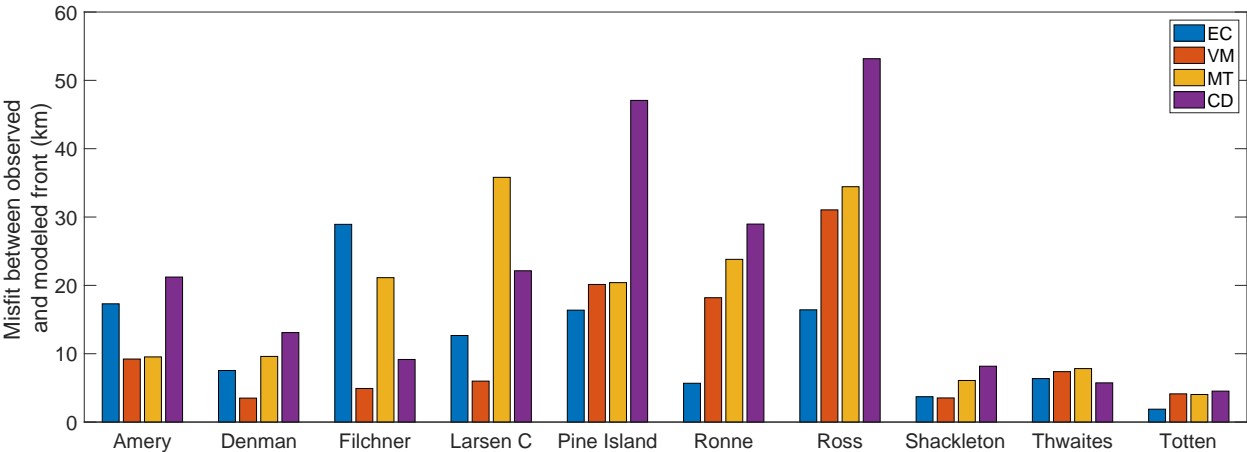

**Figure 5.** Modeled misfit distances for each calving law by ice shelf.





## 4 Discussion

Given the results of Fig. 5, it is instructive to consider why EC and VM perform best for the majority of ice shelves under consideration. A first-order answer may invoke the notion that EC is, in some sense, tuned to follow embayment coastlines,

and VM is tuned to ice velocity profiles. This occurs on account of their formulations, with EC increasing calving rate when ice exits an embayment (by virtue of across-flow strain rate increasing; see Eq. 1) and VM increasing calving rate where velocity increases (see Eq. 2). As a consequence, ice shelves whose fronts already follow coastlines may tend to favor EC, while ice shelves with more complex velocity profiles may be "evened out" by VM. By contrast, MT and CD are highly dependent on local conditions, particularly ice thickness, that may vary on small spatial scales. This may render MT and CD unlikely to

consistently capture front-wide calving processes that operate on larger spatial scales.

Taking EC and VM to be the most effective calving laws of the four considered, we next investigate the possible correlates that predict EC success versus VM success for any given ice shelf. Performing two-sample t-tests (with the important caveats that data are not normally distributed and sample size is exceedingly small, with $n = 9$), we find no significant differences in mean frontal thickness or mean frontal velocity between those ice shelves best characterized by EC versus VM. Another

potential factor that does not yield a significant difference is mean water depth along the front. With these factors unlikely to be predictive correlates, we turn to the role of buttressing as a possible predictor of calving law success. Buttressing is a natural avenue to explore, as previous work has implied a relationship between ice shelf weakening, reduced buttressing, and calving (Borstad et al., 2016). We reproduce the buttressing calculations of Fürst et al. (2016) to quantify buttressing along the second principal stress direction ("maximum buttressing", in keeping with the existing nomenclature) and along the flow direction

("flow buttressing") over the entire floating extent of the ice shelf. As in Fürst et al. (2016), we compute a normal buttressing number $K_n$, such that

$$K_n = 1 - \frac{\boldsymbol{n} \cdot \sigma \cdot \boldsymbol{n}}{N_0}, \tag{12}$$

where $\boldsymbol{n}$ is a unit vector either in the direction of the second principal stress (for maximum buttressing) or the flow direction (for flow buttressing), $\sigma$ is the resistive stress tensor, and $N_0$ is the vertically integrated pressure exerted by the ocean on the

ice front:

$$N_0 = \frac{1}{2}\rho_i \left(1 - \frac{\rho_i}{\rho_w}\right) gh. \tag{13}$$

Computing mean values of maximum buttressing and flow buttressing along ice shelf fronts, we again find no significant differences between ice shelves best characterized by EC versus those best characterized by VM. Also computing the mean value of maximum buttressing along the grounding line yields no significant difference.

To take a different approach, we next compute the mean buttressing value over the entire ice shelf rather than along the front or grounding line. We use this value to calculate the proportion of "passive shelf ice" (PSI), defined by Fürst et al. (2016) as those regions of shelf ice which have little or no dynamical influence; in other words, PSI does not significantly alter the flow or stress regime if removed. We set different thresholds for computing PSI area based on maximum buttressing values, ranging from 0.1 to 1 at intervals of 0.1. From here, we calculate the percentage of the entire shelf containing PSI according to each





buttressing threshold, and compare EC- and VM-best ice shelves. We find that, across all PSI thresholds, ice shelves fit best by VM nominally have a greater proportion of PSI than those fit best by EC (Fig. 6). This apparent pattern implies that EC is better tuned to buttressed embayments, and VM is more robust to unbuttressed ice extending outside of embayment margins. Note that buttressing can exceed 1 when opposing ocean pressure and lateral confining pressure exceed the ice flow driving force, in which case ice is in a highly compressive regime. Similarly, the buttressing value can be less than zero when ice is in

a purely extensive stress regime (Fürst et al., 2016).

To illustrate how EC and VM may exploit the same buttressing regime in different ways, consider Fig. 7. Here, we show the evolution of buttressing for floating ice over the 200-year model simulation. For Amery Ice Shelf, VM maintains the original unbuttressed extent outside of the embayment (Figs. 7a and 7g), while EC removes the unbuttressed extent outside of the embayment and trends roughly towards the embayment cutoff (Figs. 7a and 7d). For Ross and Ronne Ice Shelves, VM allows

unbuttressed ice to advance beyond the embayment coastline (Figs. 7h and 7i, respectively). By contrast, EC maintains ice at approximately the extent of the embayment coastline (Figs. 7e and 7f), with some thin ice tongues advancing beyond the embayment coastline in regions of localized low across-flow strain rate $\dot{\epsilon}_\perp$. These observations support the notion that, upon exiting embayments, VM facilitates advance in unbuttressed regions and EC trends towards the embayment cutoff, removing unbuttressed ice if present. This is consistent with the idea that EC calves ice at a faster rate when across-flow strain rate $\dot{\epsilon}_\perp$

increases, i.e. upon exiting an embayment and removing the influence of embayment margin stresses. This pattern does not hold for all ice shelves under consideration (namely, Larsen C, where we observe an opposite trend, possibly due to the influence of topographic pinning points such as Gipps Ice Rise), but it represents a simple mechanistic explanation that is consistent with calving law theory.

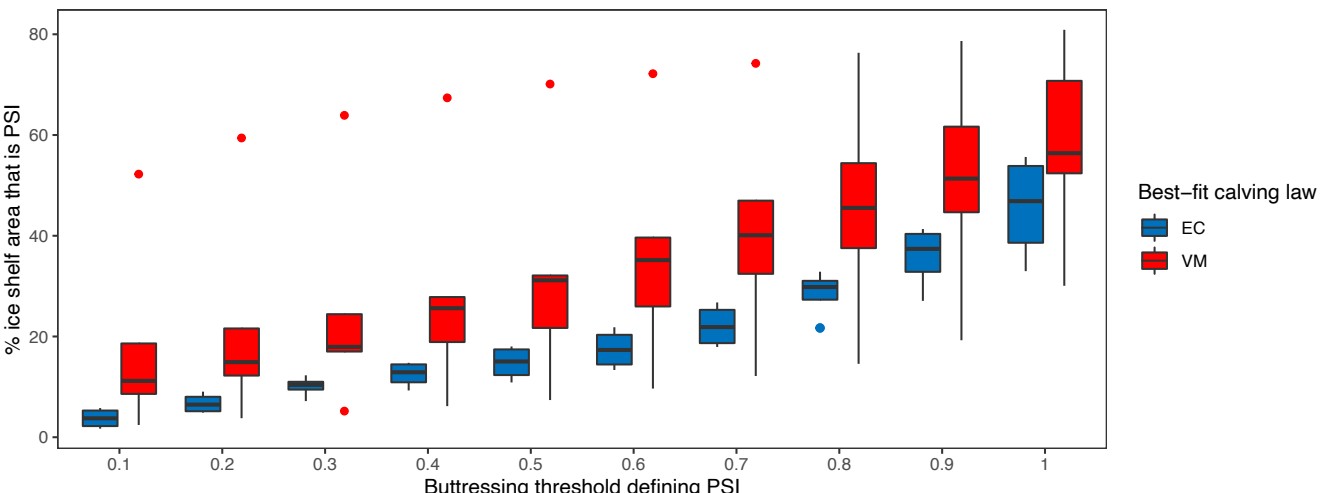

**Figure 6.** Box-and-whisker plot grouped by best-fit calving law (EC or VM) for different buttressing thresholds defining PSI plotted against the percentage of shelf area that is PSI. Dots represent outliers, with the upper VM outlier being Shackleton Ice Shelf.





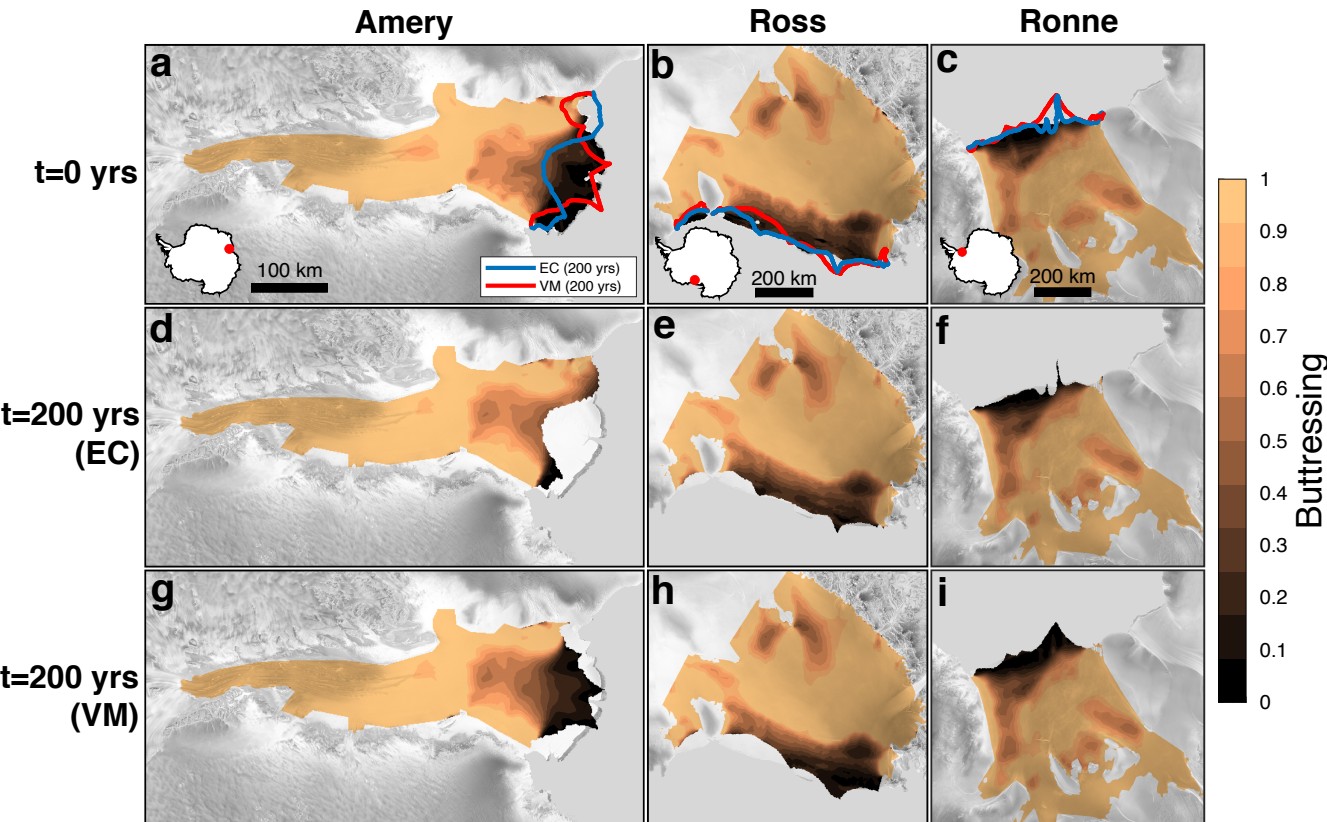

**Figure 7.** Three ice shelves (Amery, Ross, and Ronne) where the best-fit EC model run removes most unbuttressed ice outside of the embayment **(d-f)** and where the best-fit VM model run maintains an unbuttressed ice extent outside of the embayment **(g-i)**. Here, only floating ice is plotted in colors. Ice shelf location is shown by a red dot on an Antarctica inset. Model results are overlaid with a radar image in grey.

Based on these results, we recommend that EC and VM be used on a case-by-case basis for modeling Antarctic ice shelves
on centennial timescales. However, as climate continues to warm and ice shelves likely retreat deeper within their embayments, across-flow strain rate $\dot{\epsilon}_\perp$ may decrease in regions where the ice flow is channelized through deep troughs with parallel side walls, rendering EC intractable as a long-term solution given that it will consistently readvance the ice front to the embayment coastline under these conditions. Thus, as Antarctic ice shelves become more akin to fjord-confined Greenland glaciers in that they contain a higher proportion of buttressed ice, VM may become a more appropriate choice, as in Choi et al. (2018) for
Greenland. This recommendation is not inconsistent with the results of Fig. 6, as Fig. 6 compares against the modern-day scenario where ice fronts generally exist at or downstream of the embayment margin instead of within the embayment. To this end, we also recommend that future numerical experiments investigate how calving will occur as ice shelves retreat upstream of coastlines, where ice is more likely to be highly buttressed. To date, the collapse of Larsen B Ice Shelf is the only such observation of this process occurring, so modeling efforts represent our best opportunity to learn more about this process.





Although simple to conceptualize, our misfit metric involves some important caveats. Most notable is its dependence on observed calving front length. Meandering calving fronts, which have greater lengths than straight calving fronts spanning the same embayment width, will result in a smaller misfit metric since the calving front length is in the denominator. This is true so long as the the misfit area in the numerator does not increase commensurately. For this reason, it is not particularly valuable to compare absolute misfit metric values between different ice shelves which have dissimilar observed calving front lengths.

Instead, it is more informative to compare the relative order by which calving laws minimize the misfit metric, as misfit is consistent between each calving law for a given ice shelf. In this sense, calving law order is comparable between ice shelves. Another limitation of our misfit metric is its inability to distinguish between regions of advance and regions of retreat for a given model run. Retreat and advance contribute equally to the misfit calculation since the area between the the two fronts is unsigned. While most models either mostly retreat or mostly advance (or remain in steady state) for a given tuning parameter

value, there are some exceptions (e.g., the best-fit CD run for Larsen C) that preclude easy interpretation. As an alternative to our misfit metric, we computed the Fréchet distance (e.g., Alt and Godau, 1995), a widely used measure of curve similarity, between the modeled and observed fronts. The Fréchet distance may be defined as the greatest distance between ordered pairs of points on two lines. In general, we find that the Fréchet distance closely replicates the results of our misfit metric, such that the calving law which minimizes the misfit is broadly the same for our metric and the Fréchet distance.

## 5   Conclusions

We evaluate and compare four calving laws for ten ice shelves in Antarctica, relying on an assumption that Antarctic ice shelves tend to maintain a steady state calving front position over decadal timescales. We find that the majority (9 of 10) of modern observed ice shelf fronts are best fit by either the eigencalving (EC) law or the von Mises (VM) law for 200-year 2-D SSA model runs in ISSM. The minimum thickness (MT) law and the crevasse depth (CD) law are largely unsuccessful by our misfit

metric. Nominally, the proportion of passive shelf ice (PSI) is a predictor of EC versus VM success for a given ice shelf. This reflects the fact that ice spreads laterally upon exiting an embayment and thus increases in lateral strain rate, prompting the EC law to increase calving rate and limit the ice front to the embayment coastline. No such mechanism occurs for VM, so we suggest that VM is often a better choice for ice shelves that do not terminate at their embayment coastlines. Future work should consider more complex calving laws, such as those incorporating damage criteria or rifting physics, and should address

the nature of calving as inevitable retreat of shelf ice into the highly buttressed interiors of embayments occurs. Additionally, once the observational record becomes sufficiently long (∼50 years), supplementary studies should replicate this work without our assumption of a steady state calving front position.

*Code and data availability.*   ISSM is open source and is available at https://issm.jpl.nasa.gov (Version 4.22, last access: June 7, 2023). The initial geometry is from BedMachine v3 (Morlighem et al., 2020) and is available at https://nsidc.org/data/nsidc-0756/versions/3. The code

of the simulation and data analyses is available at https://doi.org/10.5281/zenodo.8015164.



*Author contributions.* JW and MM designed the setup of the numerical experiments. JW conducted the numerical experiments and data analysis, with assistance from MM and GC. JW wrote the first version of the manuscript, with input from MM and GC.

*Competing interests.* The authors declare that they have no conflict of interest.

*Acknowledgements.* JW and MM acknowledge support from the National Science Foundation (award no. 2147601), "Collaborative Re-
search: Frameworks: Convergence of Bayesian inverse methods and scientific machine learning in Earth system models through universal differentiable programming." GC is supported by two Heising-Simons Foundation grants (no. 2019-1161 and no. 2021-3059).



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
