# Peer review of "Evaluation of four calving laws for Antarctic ice shelves"

_The Cryosphere, 2023_

## Referee Comment (RC1)

Summary and comments on the manuscript
tc-2023-86 entitled

**Evaluation of four calving laws for Antarctic ice shelves**

presented on 08.06.2023
by

J.A. Wilner et al.

**SUMMARY**

With this manuscript, the authors aim to put the applicability of
four calving laws to the test for ten ice-shelf settings in Antarctica.
For this purpose, they employ a state-of-the-art ice-flow model,
which comprises accurate calving front tracking.  For each ice shelf,
the model is initialised with present-day maps of ice geometry and
surface velocities.  Consecutive forward simulations are undertaken
for 200 years with constant climatic conditions.  Assuming that
present-day ice-front positions are in steady state, the authors
then compare the final geometry with the observed calving-front
positions.  The areal mismatch serves as a quality measure.  Results
suggest that the performance of the *von Mises* laws and the *eigen-calving*
are comparable.  The former seem superior with regard to a buttressing
analysis, because are larger fraction of passive shelf ice (PSI)
is preserved - more comparable to observations.

The manuscript is very well written and illustrated and therefore
easy to follow.  The authors also formulate very concise objectives.
However, I miss some important details of the experimental setup,
which might well have implications for the interpretation of the
result.  Do not misunderstand me, I remain very positive about this
manuscript and I recommend that the editor should continue to considered
it for publication in *The Cryosphere* after my concerns have been
alleviated.  These concerns certainly imply a major revision.

**MAJOR COMMENTS**

**EXPERIMENTAL SETUP**
The introductory paragraph to section 2 (L102-117) serves to explain
the experimental setup.  Yet key information is missing here.  As
I understand it, your model domain only comprises the floating part

of the benchmark ice shelves.  I immediately wonder about upstream
boundary conditions at the grounding line.  I suspect observed velocities.
For reproducibility, please specify the time period of the constant
RACMO forcing.  Finally, I wonder about your treatment of sub-shelf
melting.  This component is key to keep the ice-shelf geometry in
balance/close to present-day.  Yet the basal mass balance is unspecified.
Please amend.

**INITIALISATION**
I understand that inverse techniques are used to get an initial
model state from observations on ice geometry and surface velocities.
After such an initialisation, there is no guarantee that subsequent
simulations do not strongly drift away from these initial states
(spurious flux-divergence, etc.).  Even if such simulations would
equilibrate after 200 years, the ice-shelf settings might be very
different.  Unfortunately the authors show no temporal evolution
or quantify the overall mismatch between observed and modelled quantities
of ice velocities, thickness or grounding line positions.  Did you
check if the volume evolution actually equilibrated or is there
still a drift or some oscillations after 200 years.  An additional
equilibration simulation with a stationary calving front position,
prescribed from observations, could prove useful.  In absence of
such reference information, it is very difficult to assess your
calving-law analysis.

**BUTTRESSING**
I did enjoy reading the sections on buttressing and more specifically
on the passive shelf-ice analysis.  I consider it an interesting
addition.  Yet I am not convinced about your decision to presented
it in the discussion section.  Please consider transferring it to
the results.  Moreover, you are certainly aware that this analysis
stays highly qualitative.  In a way, I think it mostly boils down
again to your areal analysis.  Admittedly, there are some differences
in the butteressing fields between the various steady-state geometries
(Fig.  7) – also some distance usptream of the calving front.  This
difference might, however, be indicative for important differences
in ice geometry and/or velocity impeding a clean comparison.  Please
check.  Finally, the locally derived buttressing number from Fürst
et al.  (2016) has been challenged as an appropriate indicator for
grounding-line buttressing (e.g., Zhang et al., 2020).  You should
pick this up in your discussion.

**DISCUSSION**
You start your discussion by a qualitative assessment of the EC

and VM formulations and consequences on the expected frontal shapes.
I am not sure if I fully follow your argument for the 'evened out'
VM calving fronts (L258). In Fig. 3b and 4c, the VM law results
in some elongated ice-shelf protrusions. In a latter section (L304-314),
you expand this assessment to the anticipated frontal retreat under
climate warming, which might less well be captured by EC. This is
nicely substantiated by the results from Choi et al. (2018) on
Greenland. I would transfer this section to the beginning of the
discussion - just after your first explanations. I also appreciate
your discussion of the computation of the areal mismatch - it is
valuable. However, I completely miss any assessment of the experimental
setup, involving model initialisation, calibration and equilibration.
I wonder if you can compare the calibrated parameters (Table 1)
to realistic ranges or values from other studies. As climatic conditions
over Antarctica are not in steady state, also the assumption that
the observed calving front should be reproduced in your model setup
can be challenged. Please extend your discussion.

**MINOR COMMENTS**

**L1** Insert 'for each ice-shelf setup' after '[...] of these calving
laws'.
**L222-224** Does your model see a pinning point beneath the Eastern
section of the Thwaites Ice Shelf? If not, you might want to introduce
some friction there. It would be good to include a supplementary
figure on how modelled and observed velocity fields (and/or geometries
along a flow-line) compare right after the inversions. Best for
all ice-shelves.

FIGURES
**Fig. 1 - 4** Can you specify if you show modelled or observed velocities
as background field. I guess that Fig. 1 shows modelled results,
while the others show observations. Which radar image (from which
mission/sensor) is shown as grey shading.

**Fig. 6** As it stands, this figure allows us only to compare PSI
fractions for two calving laws. To me, it seems not possible to
judge, which one is more reliable. I therefore suggest that you
add the PSI fraction right after the model initialisation (t=0yr).
This is not difficult and would give a baseline for comparison.
As this plot aggregates information from all ten ice-shelves, I

suggest that you further add a supplementary figure, which presents this PSI analysis for each ice-shelf. This might substantiate your argument why to prefer VM over EC.

**REFERENCES**

Fürst, J. J., Durand, G., Gillet-Chaulet, F., Tavard, L., Rankl, M., Braun, M., and Gagliardini, O.: The safety band of Antarctic ice shelves, Nature Climate Change, 6, 479, 2016.

Choi, Y., Morlighem, M., Wood, M., and Bondzio, J. H.: Comparison of four calving laws to model Greenland outlet glaciers, The385 Cryosphere, 12, 3735{3746, 2018.

Zhang, T., Price, S. F., Hoffman, M. J., Perego, M., and Asay-Davis, X.: Diagnosing the sensitivity of grounding-line flux to changes in sub-ice-shelf melting, The Cryosphere, 14, 3407{3424, 2020.

---

## Referee Comment (RC2)

**Review of:**
**"Evaluation of four calving laws for Antarctic ice shelves" by Wilner et al**

Wilner and colleagues assess how accurately four calving laws represent ten Antarctic ice shelves under the assumption that the ice front of these ice shelves are in steady state. For this purpose, they use a state-of-the art ice sheet model run at high resolution in constrained Antarctic domains. The metric they use to quantify the realism of the calving laws is based on the areal mismatch. Their results show that the eigencalving law (EC) and the von Mises law (VM) are the ones which best reproduce the calving front position. A further analysis based on the passive shelf ice (PSI) suggests that the VM law represents more accurately the observed PSI computed by Fürst et al., (2016).

This manuscript is very well written and illustrated and it is very well suited for the scope of The Cryosphere. I do not have major concerns but I think some initialisation steps should be clarified for the reader. I also have some questions and suggestions for the authors which they may consider or not.

**General comments:**

**Initialization**

I agree with the other reviewer that some important information is lacking in the methodology, mainly if basal melting is considered and how you treat grounded ice in your ice-sheet model. Is the grounding line fixed or is it allowed to evolve?

**Ice shelf rigidity**

You say that you invert for rigidity in ice shelves. Which parameter are you tuning there? The ice viscosity parameter B? A viscosity enhancement factor for ice shelves?

**Stationary calving front position**

As suggested by the other reviewer you could do an additional equilibration simulation with a stationary calving front position fixed to observations. You could compare there for instance the calving rate at the ice front for different calving laws with Rignot et al., (2013).

**von Mises calving law**

One of the key messages of this manuscript is that the VM law best reproduces the observed calving front positions. However, as you state in the manuscript, this result can be partially explained by the fact that you invert for rigidity which is explicitly considered in the VM computation (Eq. 3). There exist other approaches in the literature for tuning ice shelves, for instance through enhancement factor (Surawy-Stepney, 2023) or basal-melting rates (Lipscomb et al., 2021), though the latter are applied to match observed ice thickness rather than velocities. Do you think that if you would have adopted another inversion method you would still have such a good ice front position with VM?

**Calibration parameter of von Mises calving law**

You calibration parameter in the VM calving law is the tensile stress threshold $\sigma_{max}$. This threshold should represent a physical property of the ice, mainly the ice tensile strength (~0.7 MPa; Morlighem et al., 2016, Bassis et al., 2021). Your obtained calibration values are lower, but in the same order of magnitude. Do you have an explanation or interpretation for this?

**Technical questions:**

- It is not clear to me how you apply calving in your ice-sheet model. Is the calving rate a thinning rate applied to the ice front or do you trace the ice front position via a level-set method?

- The crevasse depth law (CD) is only computed at the ice front or are crevasses computed over the whole ice shelf? Do crevasses affect your ice dynamics?

- How well do you simulate ice thickness with observations?

**References:**

- Fürst, J., Durand, G., Gillet-Chaulet, F. *et al.* The safety band of Antarctic ice shelves. *Nature Clim Change* 6, 479–482 (2016), DOI: 10.1038/nclimate2912.

- Rignot, Eric, et al. "Ice-shelf melting around Antarctica." *Science* 341.6143 (2013): 266-270, DOI: 10.1126/science.1235798.

- Surawy-Stepney, T., Hogg, A.E., Cornford, S.L. *et al.* Episodic dynamic change linked to damage on the Thwaites Glacier Ice Tongue. *Nat. Geosci.* 16, 37–43 (2023), DOI: 10.1038/s41561-022-01097-9.

- Lipscomb, William H., et al. "ISMIP6-based projections of ocean-forced Antarctic Ice Sheet evolution using the Community Ice Sheet Model." *The Cryosphere* 15.2 (2021): 633-661, DOI: 10.5194/tc-15-633-2021.

- Morlighem, Mathieu, et al. "Modeling of Store Gletscher's calving dynamics, West Greenland, in response to ocean thermal forcing." *Geophysical Research Letters* 43.6 (2016), DOI: 10.1002/2016GL067695.

- Bassis, J. N., et al. "Transition to marine ice cliff instability controlled by ice thickness gradients and velocity." *Science* 372.6548 (2021): 1342-1344, DOI: 10.1126/science.abf6271.

---

## Author Comment (AC1)

**Response to Reviewer #1**

J.A. Wilner et al.

August 11, 2023

**SUMMARY**

With this manuscript, the authors aim to put the applicability of four calving laws to the test for ten ice-shelf settings in Antarctica. For this purpose, they employ a state-of-the-art ice-flow model, which comprises accurate calving front tracking. For each ice shelf, the model is initialised with present-day maps of ice geometry and surface velocities. Consecutive forward simulations are undertaken for 200 years with constant climatic conditions. Assuming that present-day ice-front positions are in steady state, the authors then compare the final geometry with the observed calving-front positions. The areal mismatch serves as a quality measure. Results suggest that the performance of the von Mises laws and the eigen-calving are comparable. The former seem superior with regard to a buttressing analysis, because are larger fraction of passive shelf ice (PSI) is preserved - more comparable to observations.

The manuscript is very well written and illustrated and therefore easy to follow. The authors also formulate very concise objectives. However, I miss some important details of the experimental setup, which might well have implications for the interpretation of the result. Do not misunderstand me, I remain very positive about this manuscript and I recommend that the editor should continue to considered it for publication in The Cryosphere after my concerns have been alleviated. These concerns certainly imply a major revision.

**Response:** We thank the reviewer for the positive and constructive comments. The feedback in this review has improved the quality of the manuscript. Please see our responses to individual comments below.

**MAJOR COMMENTS**

**EXPERIMENTAL SETUP**

The introductory paragraph to section 2 (L102-117) serves to explain the experimental setup. Yet key information is missing here. As I understand it, your model domain only comprises the floating part of the benchmark ice shelves. I immediately wonder about upstream boundary conditions at the grounding

line. I suspect observed velocities. For reproducibility, please specify the time period of the constant RACMO forcing. Finally, I wonder about your treatment of sub-shelf melting. This component is key to keep the ice-shelf geometry in balance/close to present-day. Yet the basal mass balance is unspecified. Please amend.

**Response:** The model domain comprises the floating part of the ice shelves under consideration and, in some cases, a small portion of the adjacent grounded ice; this will now be clarified in the manuscript. Upstream boundary conditions at the grounding line indeed utilize observed velocities with fixed ice thickness, which we will now include in the manuscript. We will also clarify that the time period of the constant RACMO2.3 forcing is averaged over 1979-2011, as well as the fact that sub-shelf melting is constrained by the dataset of Rignot et al. (2013).

**INITIALISATION**

I understand that inverse techniques are used to get an initial model state from observations on ice geometry and surface velocities. After such an initialisation, there is no guarantee that subsequent simulations do not strongly drift away from these initial states (spurious flux-divergence, etc.). Even if such simulations would equilibrate after 200 years, the ice-shelf settings might be very different. Unfortunately the authors show no temporal evolution or quantify the overall mismatch between observed and modelled quantities of ice velocities, thickness or grounding line positions. Did you check if the volume evolution actually equilibrated or is there still a drift or some oscillations after 200 years. An additional equilibration simulation with a stationary calving front position, prescribed from observations, could prove useful. In absence of such reference information, it is very difficult to assess your calving-law analysis.

**Response:** These are excellent suggestions and we thank the referee for making them. A simulation of Ross Ice Shelf with a fixed calving front position (see Fig. 1 shown below) indicates marginal drift in floating ice volume ($<1\%$) and mean floating ice thickness ($\sim 1\%$) over 200 years, with somewhat larger drift in mean floating ice velocity ($\sim -4.5\%$). The larger drift in ice velocity is likely due to the shock of initial conditions. We will include a supplementary figure showing these drifts for all benchmark ice shelves with fixed calving front position applied over the 200 year simulation period. Grounding line position is largely unchanged over the 200 year simulation for all benchmark ice shelves.

**BUTTRESSING**

I did enjoy reading the sections on buttressing and more specifically on the passive shelf-ice analysis. I consider it an interesting addition. Yet I am not convinced about your decision to presented it in the discussion section. Please consider transferring it to the results. Moreover, you are certainly aware that

[Figure]

Fig. 1: Drift of various ice parameters over a 200-year simulation for Ross Ice Shelf with a fixed calving front. Only the floating portion of the domain is considered here.

this analysis stays highly qualitative. In a way, I think it mostly boils down again to your areal analysis. Admittedly, there are some differences in the butteressing fields between the various steady-state geometries (Fig. 7) - also some distance usptream of the calving front. This difference might, however, be indicative for important differences in ice geometry and/or velocity impeding a clean comparison. Please check. Finally, the locally derived buttressing number from Furst et al. (2016) has been challenged as an appropriate indicator for grounding-line buttressing (e.g., Zhang et al., 2020). You should pick this up in your discussion.

**Response:** We are pleased that you found the sections on buttressing, particularly the passive shelf-ice analysis, to be of interest. While we understand your suggestion to place this analysis in the results section rather than the discussion, we made a deliberate decision to include it in the discussion for reasons that we believe enhance the overall context and logical flow of the paper. Chiefly among these reasons is that we consider it prudent to cleanly distinguish between the

key point of the paper (the areal mismatch analysis on a shelf-by-shelf basis) and any supplementary analyses, such as the buttressing analysis, which might derive from the areal mismatch results. We agree with the reviewer's assessment of the qualitative nature of our buttressing analysis, and deem it the most concise manner in which to present these results - future work should consider more quantitative avenues. Although the subtle differences in the buttressing fields between EC and VM in Figure 7 may be indicative of upstream influence of different frontal geometries or ice velocities, we do not consider the comparison especially problematic; first, the buttressing field differences are minor compared to the differences in frontal position and second, our intent is for the reader to mainly consider the frontal positions rather than upstream buttressing effects. We will add a reference to Zhang et al. (2020) in the discussion to clarify that Fürst et al. (2016) is one way of measuring buttressing, but it strongly depends on the direction chosen at any given point.

**DISCUSSION**

You start your discussion by a qualitative assessment of the EC and VM formulations and consequences on the expected frontal shapes. I am not sure if I fully follow your argument for the 'evened out' VM calving fronts (L258). In Fig. 3b and 4c, the VM law results in some elongated ice-shelf protrusions. In a latter section (L304-314), you expand this assessment to the anticipated frontal retreat under climate warming, which might less well be captured by EC. This is nicely substantiated by the results from Choi et al. (2018) on Greenland. I would transfer this section to the beginning of the discussion - just after your first explanations. I also appreciate your discussion of the computation of the areal mismatch - it is valuable. However, I completely miss any assessment of the experimental setup, involving model initialisation, calibration and equilibration. I wonder if you can compare the calibrated parameters (Table 1) to realistic ranges or values from other studies. As climatic conditions over Antarctica are not in steady state, also the assumption that the observed calving front should be reproduced in your model setup can be challenged. Please extend your discussion.

**Response:** We thank the reviewer for these comments. In retrospect, we agree that our argument for "evened out" calving fronts is a bit convoluted (and not always accurate) as presented in the text and will clarify accordingly, or perhaps remove entirely. Our discussion about the anticipated frontal retreat under a warming climate follows from the buttressing discussion, hence its placement towards the end of the discussion section, but we may also introduce the concept towards the beginning of the discussion section as suggested by the reviewer. Regarding the points about the experimental setup, please refer to our response to the 'Initialisation' section. The reviewer brings up a good point about comparing calibrated parameter values to those of other studies, and we will include such comparative values (as available) in our revised discussion. Admittedly, the assumption of steady state climatic conditions may indeed be challenged,

and we shall extend our discussion to better account for our reasoning for this assumption.

**MINOR COMMENTS**

- L1 Insert 'for each ice-shelf setup' after '[...] of these calving laws'.
  **Response:** Changed

- L222-224 Does your model see a pinning point beneath the Eastern section of the Thwaites Ice Shelf? If not, you might want to introduce some friction there. It would be good to include a supplementary figure on how modelled and observed velocity fields (and/or geometries along a flow-line) compare right after the inversions. Best for all ice-shelves.
  **Response:** Thank you for bringing this to our attention; yes, a pinning point beneath the eastern section of the Thwaites Ice Shelf is evident in our model with the bathymetric data used here (BedMachineV3).

- Fig. 1 - 4 Can you specify if you show modelled or observed velocities as background field. I guess that Fig. 1 shows modelled results, while the others show observations. Which radar image (from which mission/sensor) is shown as grey shading.
  **Response:** We will correct the figure captions to specify whether it is the modeled or observed velocities in the background field. Radar data is from the Radarsat Antarctic Mapping Project (RAMP) Antarctic Mapping Mission (AMM-1), and we will specify this in the first relevant caption.

- Fig. 6 As it stands, this figure allows us only to compare PSI fractions for two calving laws. To me, it seems not possible to judge, which one is more reliable. I therefore suggest that you add the PSI fraction right after the model initialisation (t=0yr). This is not difficult and would give a baseline for comparison. As this plot aggregates information from all ten ice-shelves, I suggest that you further add a supplementary figure, which presents this PSI analysis for each ice-shelf. This might substantiate your argument why to prefer VM over EC.
  **Response:** These are valid suggestions, and we will modify Fig. 6 to also specify the PSI fraction at t=0 yr as well as include a supplementary figure showing the same PSI analysis for each ice shelf.

---

## Author Comment (AC2)

**Response to Reviewer #2**

J.A. Wilner et al.

August 11, 2023

**SUMMARY**

Wilner and colleagues assess how accurately four calving laws represent ten Antarctic ice shelves under the assumption that the ice front of these ice shelves are in steady state. For this purpose, they use a state-of-the art ice sheet model run at high resolution in constrained Antarctic domains. The metric they use to quantify the realism of the calving laws is based on the areal mismatch. Their results show that the eigencalving law (EC) and the von Mises law (VM) are the ones which best reproduce the calving front position. A further analysis based on the passive shelf ice (PSI) suggests that the VM law represents more accurately the observed PSI computed by Fürst et al., (2016). This manuscript is very well written and illustrated and it is very well suited for the scope of The Cryosphere. I do not have major concerns but I think some initialisation steps should be clarified for the reader. I also have some questions and suggestions for the authors which they may consider or not.

**Response:** We express our gratitude to the reviewer for the positive words and insightful feedback. Our responses to specific comments are presented below for your reference.

**General comments**

**Initialization**

I agree with the other reviewer that some important information is lacking in the methodology, mainly if basal melting is considered and how you treat grounded ice in your ice-sheet model. Is the grounding line fixed or is it allowed to evolve?

**Response:** As addressed in our response to the other reviewer, basal melting is incorporated via the dataset of Rignot et al. (2013). Some grounded ice adjacent to the floating ice is included in the overall model domain. Although the grounding line is allowed to evolve (which we will now specify in the manuscript), grounding line position is largely unchanged over the 200 year simulation for all benchmark ice shelves.

**Ice shelf rigidity**

You say that you invert for rigidity in ice shelves. Which parameter are you tuning there? The ice viscosity parameter B? A viscosity enhancement factor for ice shelves?

**Response:** We tune the ice viscosity parameter B, which we will now specify in the manuscript.

**Stationary calving front position**

As suggested by the other reviewer you could do an additional equilibration simulation with a stationary calving front position fixed to observations. You could compare there for instance the calving rate at the ice front for different calving laws with Rignot et al., (2013).

**Response:** Please refer to our response to the other reviewer in which we provide an equilibration simulation with a stationary calving front for Ross Ice Shelf. We will include a supplementary figure showing such equilibration results for each ice shelf.

**von Mises calving law**

One of the key messages of this manuscript is that the VM law best reproduces the observed calving front positions. However, as you state in the manuscript, this result can be partially explained by the fact that you invert for rigidity which is explicitly considered in the VM computation (Eq. 3). There exist other approaches in the literature for tuning ice shelves, for instance through enhancement factor (Surawy-Stepney, 2023) or basal-melting rates (Lipscomb et al., 2021), though the latter are applied to match observed ice thickness rather than velocities. Do you think that if you would have adopted another inversion method you would still have such a good ice front position with VM?

**Response:** Because $\sigma_{max}$ is tuned, changing B would lead to a change in $\sigma_{max}$ in order to get the same calving rate. Inverting for the enhancement factor would have a similar effect as inverting for $B$, or for the rate factor $A$. Regarding the question about adopting a different inversion method, it is indeed an interesting avenue for future research. Exploring alternative inversion techniques could provide valuable insights into the robustness of the VM law's performance in reproducing ice front positions, as well as the effectiveness of other calving laws. We will carefully consider such a question in the revised discussion.

**Calibration parameter of von Mises calving law**

You calibration parameter in the VM calving law is the tensile stress threshold $\sigma_{\mathrm{max}}$. This threshold should represent a physical property of the ice, mainly the

ice tensile strength ( 0.7 MPa; Morlighem et al., 2016, Bassis et al., 2021). Your obtained calibration values are lower, but in the same order of magnitude. Do you have an explanation or interpretation for this?

**Response:** Thank you for bringing this to our attention. We caution that VM remains a simple parameterization and differences in calibration values between different ice masses are not entirely unexpected given such factors as damage that weakens the ice, lowering its strength. This is just one possible interpretation.

**Technical questions**

- It is not clear to me how you apply calving in your ice-sheet model. Is the calving rate a thinning rate applied to the ice front or do you trace the ice front position via a level-set method?
  **Response:** We use the level-set method, and will clarify this in the text.

- The crevasse depth law (CD) is only computed at the ice front or are crevasses computed over the whole ice shelf? Do crevasses affect your ice dynamics?
  **Response:** The CD law is computed over the entire ice shelf, but is only numerically meaningful near the ice shelf front where the zero contour of the level-set is advected. As crevasses are not explicitly included in the ice flow simulation (their hypothetical depths are implicitly calculated in Equations 5-9 based on a variety of associated parameter values), crevasses do not affect ice dynamics here.

- How well do you simulate ice thickness with observations?
  **Response:** We use observations to initialize ice thickness, and the drift in ice thickness over the course of the simulation with a fixed front will be included in the supplement.